# The Behavior of Information: A Reconsideration of Social Norms

Jennifer A. Loughmiller-Cardinal [1,*,†] and James Scott Cardinal [1,2,†]

1 Rubicon Insight Social Consulting, LLC, Westerlo, NY 12193, USA
2 Cultural Resources Survey Program, New York State Museum, Albany, NY 12230, USA
* Correspondence: chuljenn@hotmail.com
† These authors contributed equally to this work.

**Abstract:** Do social norms really matter, or are they just behavioral idiosyncrasies that become associated with a group? Social norms are generally considered as a collection of formal or informal rules, but where do these rules come from and why do we follow them? The definition for social norm varies by field of study, and how norms are established and maintained remain substantially open questions across the behavioral sciences. In reviewing the literature on social norms across multiple disciplines, we found that the common thread appears to be information. Here, we show that norms are not merely rules or strategies, but part of a more rudimentary social process for capturing and retaining information within a social network. We have found that the emergence of norms can be better explained as an efficient system of communicating, filtering, and preserving experiential information. By reconsidering social norms and institutions in terms of information, we show that they are not merely conventions that facilitate the coordination of social behavior. They are, instead, the objective of that social coordination and, potentially, of the evolutionary adaptation of sociality itself.

**Keywords:** social norms; social institutions; information; social cognition; normative belief; cultural evolution

## 1. Introduction

We are reminded of the famous monograph in which Kroeber and Kluckhohn [1] enumerated more than one hundred and sixty distinct definitions for the term *culture*—a central concept for anthropology and sociology. The work illustrated just how densely packed and theoretically laden the term had become over time, as it was defined and redefined by generations of scholars.

The term *social norm* appears to be in a similar state through the diffusion and dilution of its meaning across multiple fields of study and their applications. There are few terms that are as pervasive, or have such fundamental significance in the social and behavioral sciences, as the concept of a social norm. Bicchieri [2,3] describes norms as constituting an underlying structural "grammar" for social interactions. Gintis [4,5] viewed norms as internalized and socially programmable preferences that constitute a correlated equilibrium towards certain beliefs, preferences, and constraints. Others view norms as self-enforcing patterns of behavior (e.g., [6]), or self-organized collective actions and strategies for coordinating group behavior [7–9]. Norms have also been described as either rules of conformity consisting of a "tool kit" of behavioral strategies (e.g., [10]) or socially constructed (yet arbitrary) regularities of behavior (see [11–14]).

Recent research on social norms has been motivated by more practical concerns related to normative social institutions with respect to questions of public health [15], environmental issues [9,16,17], economics [18], or civil society [19,20]. Other concepts of social norms are used in numerous disciplines, ranging from philosophy (e.g., [21,22]) and law [23–25] to neuroscience [26–28] and artificial intelligence [29–32].

Each of these applications need to operationalize the concept of norms somewhat differently, and are consequently defined according to the specific domain of study and research interest. Despite the considerable volume of norm-related literature, there is little consensus as to what social norms are. Legros and Cislaghi [33] (p. 62) remark that "[t]he cross-disciplinary manifestation of the social norms concept has meant the literature on what norms are and how they affect people's actions has grown in very different directions and today includes several, often conflicting, theories."

A concept of social norms is evidently a necessary construct for describing human social interaction [2], cooperation [8], and collective behavior [7], but inconsistent connotations for the concept have arisen between disciplines. A "social norm" may mean something quite different to anthropologists studying cultural evolution and transmission (e.g., [34,35]) than it would to economists (e.g., [4,7,36]) studying behavioral game theory, or to sociologists or psychologists (e.g., [37–39]) exploring systems of behavioral self-regulation.

These inconsistencies, and the ensuing ambiguity when the term is invoked in different contexts, render present accounts for the nature, origins, and influence of social norms difficult to reconcile. The methodological implications are highlighted by Fehr and Schurtenberger [8] (p. 458), noting that:

> "...without a clean empirical identification of the relevant norms almost every behaviour can be rationalized as norm driven, thus rendering norms useless as an explanatory construct. This raises the question of whether social norms are indeed causal drivers of behaviour and can convincingly explain major cooperation-related regularities."

Moreover, "social norm" has become a superordinate term—i.e., one so broad that it requires extensive explanation by the author. Wallen and Romulo [40] observe "...the general term 'social norm' is a hypernym, an explicit definition is crucial to discussions of its place and usefulness in solving complex social-environmental issues." If the definition and status of social norms is contextually ambiguous, either ontologically or operationally, it severely limits the explanatory utility of the concept. Coherent definitions for the nature and mechanisms of social norms—from origins to outcomes—need to be consistent between fields of study and their various applications.

The discrepancies and inconsistencies between various conceptualizations of social norms largely stem from competing assumptions regarding the scope and locus of social functions. Legros and Cislaghi [33] (p. 66) found that the disagreements tended to divide between two underlying premises, "[o]ne major distinction that emerged in our analysis is whether social norms are an individual or collective construct... As individual constructs, social norms are understood to be psychological states of individuals, such as beliefs or emotions. As collective constructs, they are understood to be conditions or features of social groups or structures." They further note that the choice of approach tends to correspond with the subject or objectives of research and academic discipline, and that various integrated approaches have been proposed (see [33], pp. 66–68).

As social norms are a central concept in the social and behavioral sciences, we need to reconcile these sorts of conceptual contradictions between different research interests as well as between individualist and collectivist accounts. We propose that a necessary first step towards such reconciliation is to reexamine whether it continues to be productive to view social norms as rules, strategies, or preferences. These are all natural and intuitive descriptions for what norms do and have been useful heuristics or analogies for modeling the effects and behavior of norms, but something else is needed for a sufficient theoretical generalization of what norms are and to explain why social norms appear to have such influence.

In our review of the literature, we believe a reconciliation among the various definitions exists by exploring the processes through which rules, strategies, or preferences are themselves derived. Each is predicated on some prior evaluation of socially mediated information, and each reflects some measure of collective validation and standardization of that information. In other words, the coordinating effects of social norms may be better

understood as the outcome of this process of normalizing the variability of information from collective experiences to *identify* points of equilibrium rather than *as* that equilibrium.

Our objective in this paper is not to refute or critique specific theories or methods, but to propose a conceptual framework to reconcile divergent aspects of those theories. We consider social norms and their relation to the emergence of normative and institutional entities with respect to the capture and maintenance of socially embedded information. We suggest this as a way to bridge the theoretical divides between individual and collective accounts of sociality and norms by examining the processes through which pro-social rules or strategies may be derived from the communication and collective validation of experiential information.

The following discussion explores social norms as the normalization of collective social information that represents the convergence of mutual information across a given social network. Social norms and normative institutions are the natural and necessary consequences of a fundamental need to filter and curate that information. These are, we will argue, the origins of normative social behaviors rather than their products. We present a reconsideration of the concept of social norms, their origins, and their influences based on the premise that norms constitute an effective and efficient infrastructure for curating information derived from collective experience.

## 2. (Re-)Defining the Problem of Norms

The terms *norm*, *normative*, and *institution* are typically used to describe either social or cognitive mechanisms that establish the basic expectations and ground-rules for social interactions within a group [2,11,13,14,41,42]. The prevalence of these terms and numerous related concepts in the literature of social and behavioral sciences suggest a nearly ubiquitous influence for these social mechanisms. Norms and institutions appear to be both the medium and means for social interaction and promoting societal cooperation and cohesion.

Where views diverge is in the degree to which norms are considered: conscious choices and justifications by individuals or as a group [43,44], whether they are imposed or consensual [37,45], their roles in social function and socialization [7,8], and what conditions translate between normative beliefs and performed behavior [3,16,46,47]. Reviews of social norm literature by Gintis [4], Mesoudi [48], Shulman et al. [49], and Legros and Cislaghi [33] provide useful context for the overall landscape of theoretical approaches across the various fields of study.

The study of norms and institutions has a long and distinguished intellectual history. Earlier scholarship on the subject sought to understand their moral, social, political, or philosophical connotations (e.g., [11,50–54]; also see historical overviews in [55,56]). The currently prevailing characterizations of norms are more concerned with aspects of social cognition and the social influences on individual or group behaviors, and tend to gravitate around some permutation of "social rules" and "common beliefs and preferences" (e.g., [2,3,8,11,14,42,57–59]).

These shared expectations of what ought to be done, said, or believed by members of a social group are described as prescriptive or proscriptive rules that constrain or guide a social actor's choices [2,5,60,61]. Similarly, *normative* broadly implies a social pressure through which norms are maintained, promoted, or enforced. A practice, institution, or ideal of a group is normative if it serves to regulate the acceptable limits of tolerance by which to evaluate conformity or nonconformity with the preferential norm [5,58,61–67]. Under these definitions, normative would signify the overall set of ideals that regulate norms, whereas norms would be bounded by that set of normative constraints.

We also know that human behavior and interactions are embedded in multiple overlapping social networks, institutions, and environments. That has been well-established in the social sciences, both theoretically and empirically [68–72]. The cumulative effect of interactions within these networks result in coordination between the behaviors of people within the group, which has been successfully modeled through simulations of repeated *n*-player games (e.g., [3,5,36,73–78]). Such models show that the assemblage of strategies,

beliefs, and expectations behave as a sort of correlated equilibrium [4,73,74,79], which corresponds with normative preferences towards multiple diverse strategies dependent on agent/"player" information and communication.

The combination of game theory and network models has provided valuable insights for describing various social network interactions. There remains, however, something of a disconnect between simulations of network processes and either empirically observable phenomena or the specific beliefs that arise in rational agents (e.g., [2,3,80–83]). In other words, even though we can simulate the emergence of a novel norm (e.g., [84–86]) or demonstrate how a norm might evolve in a heterogeneous social environment (e.g., [6,77,87–89]), these are still simulations of an idealized trait or set of rules and properties analogous to mental or behavioral processes and phenomena related to social norms. These are necessary descriptive models and heuristics for the related processes, but are not of themselves sufficient as explanatory models for social phenomena [90–95].

Substantially open questions remain pertaining to which rules and strategies emerge as social norms or which preferences may become ensconced as normative expectations or institutions [3,11,55,56,96,97]. We have a conceptual understanding of how novel properties such as conditional preferences, social norms, or normative expectations can emerge and evolve within a social network, but (as noted in [3,70,97–100]) we lack a framework for the antecedent conditions promoting the formation of incipient norms. This theoretical gap, between formation and emergence, limits our understanding of the causal relationships between social norms and behavior [49,62,99–103] as well as what promotes changes within (or deviations from) empirically identified norms or related behaviors [17,70,104].

The formation and causal relationship questions related to social norms present significant theoretical and methodological problems, especially towards the implementation of social norm interventions (e.g., [15,47,101,103,105–107]). Quite often, what people report that they believe they should do does not match what they actually do in observed behavior [3,65,108]. This raises additional questions regarding the theorization of norm compliance [48,109,110], such as the status and force of norm internalization (e.g., [111–116]) or the role of external enforcement (see [45,114,117–120]). Defining social institutions has proven difficult as well (e.g., [11,55,60,78,121,122]), since both the formation of and compliance with formal institutions exhibit similar causal ambiguities.

If social norms and normativity are as fundamental to social behavior as their ubiquitous usage makes them appear, then we need to reevaluate the assumptions regarding conditional social expectations, rules, and strategies.

## 2.1. The Strangeness of Normality

For the most part, people are surprisingly predictable. Our days are filled with simple routines and habits, each involving little conscious thought and all formed from a lifetime of preferences and experiences. We modify our behaviors as we move from private to public spheres or when interacting with acquaintances or strangers, and we do this without much thought or effort. There is rarely a need to dwell on why we do what we do. After all, it is just normal. On the other hand, we immediately know when something is not as it should be or if someone is behaving in an unusual way. Even without knowing specifically what is different, anomalies in other people's behavior or in our environment are noticeable. We are aware something is not normal.

How we acquire this effortless sense of what is or is not normal is a surprisingly complex question [123–125]. As commonly framed in social norms literature, that sense of "normality" is grounded in an interplay between the individual's expectations of others' behavior (e.g., *descriptive* norms as in [2,126,127]), the individual's beliefs about what others expect them to do (e.g., *injunctive* norms such as [79,128–130]), and collective beliefs regarding what any member of the group ought to do (e.g., *conventions* [131–133]). When something is normal, it is expected or anticipated. It is unsurprising, which implies some basis in prior information as well as subsequent validation of the expectation.



Norms—whether descriptive, injunctive, or conventional—can be argued to minimize the experience of surprise for both the individual and collective [123,124,134]. They establish conditions under which expectations of "normality" may be anticipated as likely. Consider, however, the vast amount of information that an individual would need to process in order to navigate this landscape of expectations. Nevertheless, we improvise and adapt constantly and efficiently to the actions and expectations of others within our social environment.

Somehow, though, social norms can be strict taboo prohibitions (e.g., [135]), conventions for relatively mundane activities or etiquette (such as [38,136]), or even common expectations of aesthetic preferences [137,138]. Certain norms can be persistent, while others appear to be fleeting [139,140]. The adherence and enforcement of norms can be highly variable—at times, certain norms may be violated without significant sanction, while other transgressions can have profound social consequences [45,47,109,114,141,142]. Norms can also be very specific and highly contextual, making a behavior appropriate in one setting and yet prohibited in another [66,140,143]. These inconsistencies—rigorous or lax, persistent or ephemeral, absolute or contingent—belie the central role that even seemingly trivial norms should play as coordinated rules for social performance.

Meanwhile, these variable norms and their consequent normative phenomena describe mechanisms for enforcing communally beneficial conformity (i.e., shared expectations of what "ought" to be done, said, or believed [28,51,61,144]). The cumulative effect of an individual's knowledge of (and adherence to) all these various expressions define what is collectively considered to be "normal" behavior. Group membership is contingent on conforming to those expectations, which then requires an individual to adapt their behavior within that context (such "signaling" is discussed often, see [6,14,37,66,144–148]).

Despite the diversity and variability of norms and normativity, they clearly serve to maintain cohesion of the group as well as to signal group membership (see, for instance [62,140,149–151]). An individual's adherence to prevailing group norms is demonstrated by self-regulating their performance within the acceptable tolerance of the group. This assumes that the cohesion of the group is both a sufficient and necessary justification for an individual's compliance [143,152,153]. It does not explain the collective utility or adaptive advantage of membership to the constituent members of the group.

The general evolutionary view is that individuals form social groups for the benefits of collective outcomes—e.g., improved chances of survival through mating, protection, or acquiring resources [44,48,154–163]. The presumption is that group members sacrifice a degree of autonomy, contribute resources, and take on the shared pool of risk in exchange for lowered individual risks and access to collective resources [73,89,133,157,164,165].

Such cooperative and fairness norms make sense for both individuals and the group from an evolutionary perspective, and a substantial body of research has demonstrated just that utility [8,77,160,166–171]. However, this same logic cannot be applied universally to all norms and normative expectations.

The value of the group's continuity as a collective primarily depends on the perceived utility of membership to the constituent individuals. This highlights a potential issue with understanding norms as either rules or strategies that act as constraints on individual action [38,53,172,173]. High-level principles of cooperative behavior may be explained by their adaptive advantages, but norms that arise to support and give practical structure to that ideal are arbitrary [13,139,148]. This creates an enormous problem for examining or understanding social norms solely in terms of adaptive utility or perceived function.

A century of research has highlighted that norms and social entities pertaining to political, economic, or moral and spiritual systems of practices take widely divergent forms of implementation. We may be able to identify that a norm relates to an advantageous adaptive strategy, but we cannot provide an explanation for how any particular implementation of that strategy emerged.

This sentiment is frequently implied, if not directly expressed, as in Ullmann-Margalit [174] (p. 6) for example:

"It is my feeling that the pragmatic aspects pertaining to the socio-psychological contexts in which norms emerge, exist, and disappear have been relatively neglected in recent and current philosophical discussions, possibly because of the aforementioned reason – that the great variety of contexts concerned precludes adequate systematization."

This becomes even more difficult when trying to understand practical or adaptive strategies relating to conventions, etiquette, or aesthetics. For an outside observer, the various norms and practices of another group may seem strange—perhaps even irrational or maladaptive—but to members of that group, it is simply normal (consider [124,140]).

One would be very hard pressed to imagine any social group without a system of norms in place. Therefore, and somewhat perplexingly, we are left with a social mechanism that is both arbitrary and intrinsic to basic social functioning, with the only functional constraint being that a norm exists.

### 2.2. Normalizing Norms

That norms appear to be both arbitrary and necessary to social behavior implies the underlying process of selection, validation, and normalization is the critical attribute of norms rather than the resulting rules or strategies. Norms clearly play a key role in social functioning, but promoting collective benefits is not a sufficient explanation. Preferred strategies would need to be proven as beneficial against alternatives before they could become a rule or standard—i.e., they must be the result of a more rudimentary process.

We propose that the underlying process in question is the normalization of mutual information. The pertinent concept is *information* rather than utility. Norms establish a shared pool of prior information on which both individual agents and the collective (as a community of agents) can draw. This allows members of a community access to a shared source of information—i.e., making "normality" a shared resource, pooling the cumulative knowledge and experience of the collective. Therefore, all members have information at hand to anticipate and react to novel or improvisational scenarios without prior direct knowledge.

It is information that is being normalized, not the specific practices or strategies. The resulting practices may still be quite variable, as may be the tolerance for variations within and between communities.

Social norms represent a coalescence of shared experiential information within an interacting cohort of individual agents. These norms comprise a pool of knowledge held by members of a community regarding the outcomes of their collective experiences. Shared information facilitates the coordination of practices, while the possession of shared knowledge is demonstrated through conforming performance. Rules are observable in performance, and acceptable performance signals are observable in the compliance of norms. To clarify our terms, practice is the application of skills and knowledge, whereas we are using performance as the formalized and empirical expression of norms.

By sharing these strategies, methods, and practices with others through social networks —merely through interaction and communication—shared information becomes refined with each success or failure experienced by members of the group. Curating the knowledge of these outcomes is instrumental in adapting and refining normalized strategies, methods, and performances. Norms and normative institutions serve to maintain and distribute collective experiential knowledge within a given cohort and to provide a mechanism by which to communicate that knowledge to subsequent generations of its constituents.

Norms, normative expectations, and normative institutions exist not to promote the cohesion of the group as a collective, but to promote the maintenance and propagation of the collective experiential information acquired through the interaction of individuals that constitute the group.

Essentially, social cohesion and the consequent coordination of behavior (i.e., its coordinated performance) become byproducts of an underlying causal process that has both

individual and communal utility—i.e., the curation of mutual information. The explanations of norms, and by extension the social constructs and actions associated with them, have inadvertently reversed the causal relationships of the processes involved. If norms represent mutual information, then social cohesion ultimately serves to support that mutual information and accumulation of experiential knowledge.

As counter-intuitive as it may seem, sociality exists to promote *norms*.

## 3. Norms as Information

All the experiences, observations, and outcomes from our physical and social environments are (at some level) assessed and translated into our individual pool of information. This information is the resource from which we accumulate knowledge, form opinions, establish beliefs, and make decisions. The unifying thread is that information from those events is being acquired, digested, evaluated, distilled, and utilized.

The question becomes how does a group of individuals independently coordinate the outcomes of individual processes such that they become collective beliefs and performances? Norms and normative institutions must somehow become codified through intentional collective actions, but the predicate beliefs must also somehow emerge organically from coordination between individual cognitive processes. This problem of collective coordination is central to open questions regarding the initial emergence and resilience of social norms and related social mechanisms.

Much of the current research on social norms and collective intentionality has approached the problem from the perspective of the individual acquisition of, or compliance with, information and norms within an existing social environment. The intersection between norms and social cognition is thereby defined by the individual's cognitive processing of the intentions, expectations, or mental states of others within the social environment (e.g., social or evolutionary psychology [2,25,38,58,59,79,131,175]). This form of explanation not only neglects the origins *of* that social environment, but also the effects of individual and collective adaptations *on* that social environment. The interactions of individual agency, in the course of strategically navigating and negotiating the social environment, is the generative process of that environment.

Another prominent line of research relates to cognitive frameworks for learning and decision processes that stem from a collaboration between cognitive and computer scientists (e.g., [27,176–180]). Most notably, cognitive search and decision processes are modeled after Bayesian estimation and information theory. The neurological basis of pattern detection and learning, decision-making, or stimulus response have been shown to be useful heuristics for modeling the human cognition and processing of environmental information [27,179,181–185]. These more recent theoretical approaches likewise focus on individual cognitive processes rather than the production of socially embedded information networks.

Conversely, research on the emergence and evolution of cooperative and social behavior looks mainly towards the collective and individual adaptive advantages of sociality rather than the individual cognitive processing of social information [89,154,186–191]. Without addressing the origin, the emergence of rules and strategies and the initialization of a social environment remain wholly ambiguous. It establishes the adaptive advantage of cooperative sociality, but not the mechanism by which that sociality is implemented or its organic qualities.

We propose instead that the nature of norms is a more rudimentary cognitive mechanism, which enables both social cognition and the evolution of sociality. If we consider social norms not as an inherited set of rules or coordinated strategies but as a derived central trend from the distribution of perceived beliefs and information, then the formation of norms represents the process by which social information is derived.

Although the social norms that are the outcomes of that process have coordinating effects, we propose that their intrinsic utility is in providing a reference datum against which social actors can reconcile and orient their own experience or belief as a form of

cross-validation. Thus, the primary purpose of norms would be to provide a framework for the rectification and retention of common nodes of information within the group.

Norms, as we will argue below, emerge as the perceived central tendency within bounded conceptual domains that are identified by relating experiences across the collective to those common nodes of information. What have been traditionally identified as norms and institutions should instead be considered the social performance, in the sense of the cultural pragmatics of Alexander [192], that promote the practices for retention and maintenance of that commonly identified information.

If we consider norms themselves to be the formation of underlying nodes of mutual information and belief, then the various categorizations of normative phenomena (e.g., as conventions, descriptive norms, or social norms) become moot. Each compartmentalized category of social interaction, we argue, is itself a consequence of the same underlying coordination of information and belief—i.e., a consequence of an underlying normalizing process, not the normative phenomenon itself.

### 3.1. Ontology of Social Information

Before we can describe an information-based framework for norms and normativity, however, we need to clarify what we mean by certain related concepts—i.e., experience, information, belief, and knowledge—as we will use them pertaining to social phenomena. Much like the concept of norm, these terms have become generalizations for suites of tangentially related ideas. To say something from experience [124,193], knowledge [194,195], or belief [196–198] all denote some state or degree of confidence in information. Furthermore, each entails subtle differences regarding the source of both the information and of that confidence.

Since our focus is on the social embedding of information, rather than the technical or metaphysical analysis of it (e.g., [14,54,108,178,199–202]), our usage of these concepts is more in reference to pragmatics and their roles in the social mediation of information (e.g., [5,144,186,203]). Each concept describes an aspect of transforming data, in the sense of input from stimuli or phenomena from the empirical world around us, into information. Whereas data an observation, information imputes structure (i.e., meaning and context) to the observation. Whether individually or collectively, transforming the data of life into information is embedded in social environments.

The process of those transformations is, in our view, what has been generally recognized as the social interplay between individual agency and collective action. Our intent is to describe the relationships between these transformations, so it is important to note that our use of the word *function* is meant to be understood in the mathematical or algorithmic sense. We are speaking of the collection of logical operators that define the relationships between sets of entities and not the teleological purpose of those entities, as is more common in the social sciences. In other words, we are speaking of its role and operations and not its purpose.

We use the term experience to refer to the product of an individual's direct observation or stimulus [197,204–207]. In effect, experience can be considered synonymous with data—the things from which information may be derived. The process through which experience becomes information relies on its contextualization with respect to other experiences and other information. That contextualization therefore requires some heuristic for association and validation to assign meaning and structure to the experience.

Whereas experience is an observation, belief is an expectation [131,196,197,202]. Although belief traditionally refers to a particular proposition or state, it is useful to consider what believing something entails. The propositional content of a belief is that something is or is not as it should be. It is an evaluative form of expectation. We will discuss the evaluative role of belief further below, but the role of belief is as an instrumental linking function with which to measure and evaluate the new from the known. Belief forms the basis of connecting one experience to previous experiences and the information derived from them, but belief is not itself the information derived from experience [196,197,202].

The evaluative function of belief [196,197,208,209] relates to the strength of association between an experience and prior information, rather than its validity as information. To know something, as opposed to a belief of it, implies a subsequent validation of that information [197,202,209]. That is, in essence, the distinction being made when we say we *believe* something as opposed to we *know* something. The former conveys commitment, whereas the latter implies certitude in the content of the information. Knowing entails confidence in the information as concrete or factual, while belief reflects confidence in the expectation irrespective of ambiguity or unknowns. For each, there is a certain implication of trust and confirmation in relationship to information and its basis.

Both knowledge and belief are predicated on experience, but each entails different processes of and bases for validation. Experiences may be informative yet remain anecdotal without sufficient validation to alter prior belief or knowledge. Therefore, to establish an observation as information requires a process of validation, whether through repetition or by external confirmation. The accessibility of external confirmation and association with experience is, consequently, the ultimate role and purpose of norms—i.e., a collective, standardized expectation based on curated prior information.

### 3.2. Individual and Collective Experience

We collect enormous amounts of data throughout our lives, all contributing to our understanding of, and responses to, our physical and social environments. A substantial portion of anthropological research is dedicated to understanding the frameworks responsible for our general behavior (e.g., reconsider the opening statement about culture, above). Both taught and learned, these enculturated frameworks act to filter all information that we encounter. This serves to reinforce our personal norms, and the social norms to which we are attached. We contend that norms, no matter the scale, are in fact the same general process of information filtering and the dynamic relationship between personal information and a population is a matter of the significance of individual actions and contributions.

Knowledge or belief derived from direct experience is the primary source of information we rely on. When dealing with novel scenarios, we intuitively assess for similarities to what we have already experienced [108,210]. Not only is it firsthand, but it is contextualized by our previous experiences, knowledge, and beliefs.

We extend our knowledge by accumulating indirect information through our network of social connections and communities. When no direct experience can be applied to a situation, we assess based on what we know indirectly from others—their experience, how they addressed it, and the outcome. This helps to mitigate deficiencies in our individual experiences. We are informed by all social connections and communities we keep, and our communities consequently assimilate information from our own communicated experiences [108].

These extended pools of outcomes, experience, choices, and actions are integrated with our individual pool of experiential data. In turn, they are incorporated into the beliefs and knowledge that comprise our understanding. Communally, we share information in numerous ways—e.g., discussions, stories, demonstrations, histories, or teaching. Information can also be shared through various nonverbal forms of communication. During any social interaction, there is some explicit or implicit sharing of information. "Telling is a social institution for the spreading of knowledge, enabling it to be possessed at second-hand" [108] (p. 596).

This system of feedback through exchange facilitates improvisation in novel scenarios. We might never have had a particular experience individually, but may have observed or heard about how others reacted to something similar. The constant access to information and unique experiences promotes organic adaptation within communities, and these communicated experiences become a shared pool of information. Propagating and preserving that shared information provides all members of the community an expansive resource by which to adapt beyond the limitations of their unique individual experiences.

*3.3. The Process of Belief*

The definition and nature of belief is an ongoing discussion [26,50,52,196,211–217], but contemporary views frame it as an implicit or explicit evaluation of truth or as a proposition or attitude that an individual has determined true or valid. As described above, though, we propose that belief is an expectation rather than a state, and relates to how we process information against previous knowledge. Viewed as an expectation, belief occurs at the interstices between what is known and unknown. It is an interpolation from expectation to observation, and a likelihood of an observation's divergence from what had previously been known. Belief is not itself a state of understanding, but the very process of creating that state from experience. From that view, belief serves a rudimentary function of filtering, sorting, and associating related information.

We mediate the world through beliefs formed from our own experiences, those of our community, and the shared pools of information with which we interact. We can adapt, improvise, and react to novel scenarios largely because our beliefs are not limited only to our own direct experiences. Experience, and the information derived from it, constitute the basic data of an individual's perception of the world. Comprehension of that experience, however, is dependent on its coherence and pertinence within the context of prior information and belief. Belief, as an expectation rather than state, provides a mechanism for contextualizing experiential data into usable information. Those data are contextualized by similarity or dissimilarity to what is either:

1. Believed to be true from prior experience and evaluation (i.e., a "belief *that*…");
2. Believed should be true based on all other available information (i.e., a "belief *in*…").

By evaluating the strength of similarity, belief allows us to infer associations and/or causal links with prior information and experience. These associations impute an explanatory rationale (i.e., why experience X relates to experience Y) that correspond expectations with outcomes. This process of classification maps the incoming data onto their associated prior domains by identifying similarities and inconsistencies between any new information and the expectations of prior belief. In doing so, belief regulates whether something falls within a prior domain of our experiences and evaluates for any analogous relationships with what we know.

Consequently, belief functions as a heuristic—i.e., a cluster of expectations that distills and embodies a specific domain of information [71,218–220]. Beliefs are established and reconfigured by synthesizing information with experience, and the heuristic is either reinforced or updated to reconcile that experience with prior information.

As experience accumulates, the subsequent corroboration or revision of belief establishes a cumulative mental topology of previously established expectations—effectively establishing a new or updated set of prior expectations. Each experience reinforces or rectifies those domains, providing structure to the overall landscape of belief and reducing the perception of uncertainty within it. The perceived validation of those expectations and, importantly, the expectation of such validation are where norms, as a social mechanism for shared information, begin to come into play.

The overall process that we have just described is very similar to that of an empirical Bayesian search, which has already become an influential heuristic in cognitive psychology and neuroscience (e.g., [164,182,183,221–226]). In a Bayesian model, individual beliefs represent the prior expectations and the information (communicated by other network contacts) represents the new data from which to generate new posterior expectations. Each new experience updates the existing (prior) belief based on the expectation or likelihood of the new information to derive the new (posterior) expectation. An individual's prior beliefs are comprised of a network of related expectations that were derived from previous experiences and socially acquired information. Each new experience and information gained updates the posterior credibility of that belief, and the new posterior expectations become the next set of priors.

## 4. Social Norms, Normativity, and Institutions

We have, until now, mainly been discussing experience and expectation from the perspective of an individual digesting information. An individual's conceptual terrain does not exist in isolation, however, but within a broader network of social and physical environments. It logically follows that everyone within that social environment is acquiring, digesting, and communicating information as well. Just as each agent forms heuristic beliefs and associated expectations, all constituent agents of the collective are engaged in the same process both independently and collectively. These processes occur within social networks through which information—specifically, belief about information—is exchanged (see [108,133,227–230]).

The dynamics of belief formation, described above, apply to both individual social actors and their collective interactions. The interactions of individual beliefs within the local social network behave much like the individual process of belief formation, with each individual contributing their perceived experiences and beliefs (see [14,231,232]). The collective process of updating prior to posterior expectations therefore becomes embedded in its own network of collective experiences and expectations. This emergent belief network, comprised of the aggregate belief states of socially linked individuals, is where individual and collective information combine as what we recognize as social norms.

### 4.1. Norm and Collective Expectations

No individual could reasonably be expected to have access to (or retain knowledge of) any and all contingencies and scenarios, especially when encountering novel or rare occurrences. The collective information available within a regularly interacting social cohort, however, provides a greatly expanded pool of experiential information to all members. We acquire much of our information and belief through this interactive process of comparing internal and external cross-validation [14,108,233–235].

An individual's beliefs are a combination of direct experience and perceptions of the experiences and beliefs of those around them [64,198,236,237]. Likewise, the aggregate of beliefs of the cohort is informed by the combined perceptions and beliefs of its members [12,237–239]. This system of drawing from the collective experience, as well as sharing and investing in collective beliefs by individuals, produces a certain self-selecting trend towards the coordination of belief within that community.

In other words, although the specific information and experiences of individual members of a community may be diverse, the collective information and experiences represent a much larger sample of observations. This pooled sample of experience will converge toward more accurate expectations than any individual's subset of experiences would allow. This sort of distributed search for valid beliefs over collective experiences again follows the Bayesian search model described above (e.g., [27,179,180,226]), in which repeated empirical observations update prior belief to generate more precise expectations. The inclusion of the collective experiences to update the individual's prior belief, and the subsequent updating of the collective belief as well, renders these posterior beliefs as shared nodes of information.

Once shared nodes of expectation are established between individuals within the group, these nodes consequently filter which experiences and perceptions are evaluated (individually and collectively) as valid and/or effective (see, for example, the effects of node centrality in networks as in [68,240–242]). As information is disseminated through the collective, individual perceptions and experiences that coincide between group members are augmented by the additional reinforcement. The reinforcement and coordination across a community amplifies any congruent aspects of that information and belief in the process. With sufficient amplification, those perceptions coalesce as a collective and normalized belief that is further reinforced within the group.

This coalescence is the initial emergence of a social norm—i.e., "emergence" in the sense of Bedau [243,244], Crutchfield [245], or Chalmers [246]—as the trend of individual expectations across the collective that relates to a common domain of information.

In short, the initial formation of a norm results from a rudimentary process of validating the predictive viability of an expectation between individual and collective experiences and beliefs. Similarly, the emergence of a norm related to that expectation is grounded in cross-validation by the group of similar expectations on that domain of information. As expectations coalesce towards a common perception of validity and utility, a norm regarding that domain emerges.

Importantly, however, it should be noted that what we are discussing here are not the probabilities that some piece of information is true, but evaluation of the probability that it should be believed to be true. There is a difference between the validation of belief and the verification of information. Although ideally the former serves to promote the latter, if a belief network encompasses some portion of false information, it can just as readily promote belief towards a false conclusion. The implications of the potential propagation of false belief networks cannot be overstated.

For social norms, we are looking at the network formed by social interactions between individuals and the beliefs that those individuals have derived from their own experience and information [247–249]. Experiences, expectations, and outcomes are communicated through these social network contacts, which in turn serve to both rectify or validate the individual beliefs as well as to provide new information for collective comparisons.

Repeated interactions within this social network lead to a normalization of those expectations across the group as the collective experiences provide a greater body of sampling information to each group member. Commonalities in the collective experiences and belief will therefore converge towards a normal probability distribution over successive interactions, driving the emergence and establishment of a social norm from the expectation with the maximum likelihood. The resulting norm, in both the social and probabilistic sense, describes the most likely expectation and variability for subsequent experiences.

### 4.2. Collective and Normative Expectations

The shared pool of knowledge within a social network extends the information available to any individual member, but the collective is no more omniscient than an individual in their experience of the world [248]. The coalescence of collective belief does not necessarily depend on its empirical validity, but only the interplay of mutual information and coincident perception of experiences (see, for example, recent work with online communities such as [250–254]). The establishment of common perceptions and practices is derived from the exchange of information within the group, its amplification by coincident belief, and the resilience of those coincident beliefs through subsequent perceptions of experiences. Coincident beliefs that remain stable through repeated processes of amplification emerge as common expectations, and the resulting norm remains stable only if those perceptions and inferences stay consistent.

Such a process is locally self-coordinating, but is also subject to errors of incomplete, spurious, or mistranslated information between group members (consider recent work done on decision matrices [255–259]). This combination of volatility and potential errors in both empirical validity and translation make localized norms highly fluid. To emerge as stable expectations and practices, the coalescence of the collective expectations underpinning emergent norms is necessary, but not sufficient. The process of transforming collective *experience* into collective *expectation* produces an initially unstable and "noisy" emergent norm. For an emergent norm to become established systemically, the belief on expectations and the collective perceptions of its domain of information need to stabilize [260,261].

Whereas the formation of a norm is a process of emergence, its establishment as an a priori or normative expectation (i.e., an expectation of an expectation) is a process of *convergence*. Specifically, each iteration of the process—from experience to belief to emergent norm—entails reevaluation and rectification of the set of associations for that domain of information. As the process is repeated, each iteration incorporates a broader set of experience against which to evaluate the information's validity and efficacy. This process

of distillation can efficiently condense collective experiences into a coherent domain of associated information [262–265].

Once a collective estimate for the posterior expectation converges to a steady state, however, those expectations take on characteristics of parametric probabilistic processes. In other words, there are expectations about those expectations—i.e., what most other people believe—and a perception that straying too far from those expectations is anomalous. Collective expectations now have something like an "average" belief and an allowable "standard deviation" from that belief. This allows both individuals and the collective to efficiently evaluate new information against those parameters, rather than reevaluate based on all contributing information.

The nature of this process promotes the filtering of spurious associations as more experiences are incorporated. Once there is an expectation of an expectation—essentially a parametric norm regarding an empirical norm—information that diverges overmuch from that normative expectation is identifiable as suspect. This filtering further drives the collective expectation towards an increasingly stable common perception of the mutual information. Once the collective perception reaches a stable state, the collective norm is established as a communal conditional expectation.

The prevailing perception exerts normative influence towards any subsequent information or practices associated with that domain, but is not deterministic. If new information continues to lie within a reasonable confidence of association, the converged and established norm exerts a conservative influence in retaining its relative prior expectations. The more tightly bounded the credibility of the converged norm (e.g., strong commonality of collective experience and/or long-standing confirmation), the stronger its normative influence. An established norm represents a baseline standard regarding expectations for experience and its associations within the related domain.

Note that this is not saying that the collective expectation supplants or supervenes the individual expectation. The convergence towards a collective norm only establishes the common perception of the most likely individual belief associated with a domain of information. This belief about belief constitutes the collective perceptions of associations between experience and the cumulative information available to the group [263,264,266–269].

Not all emergent norms will readily converge to stable states. For some situations, there could be multiple, equally viable, points of convergence leading to multiple norms attaching to the same domain. For others, such as rare or novel experiences, there could be insufficient information or commonality of experience for a consensus perception or expectation to converge. Similarly, highly variable events (i.e., ones for which experience would be highly diverse) could take significantly longer to converge to any stable expectation. Even for those that do naturally converge to a singular and stable normative state, the information and expectation attached to that domain of information are dependent on the continued viability and reinforcement of the collective expectations.

### 4.3. Normative Expectations and Curation of Information

If the ultimate role of norms is maintenance and persistence (i.e., curation) of collective information, then there must be corollary processes for retaining and disseminating that information. The process of norm emergence is the aggregation and normalization of individual perceptions of information. Similarly, the collective expectations derived from that normalization are also individually realized [263,267,270] (see also [245,261]). Normative expectations are essentially a norm regarding a norm—i.e., individual beliefs regarding collective expectations. Therefore, the curation of social information should be derived from the natural interplay of individual and collective expectations, rather than viewed as a collectively coordinated endeavor.

For a norm to exert normative influence requires an individual expectation regarding the collective expectation itself. This is, in effect, the inverse function of the emergence–convergence duality described above. Normative expectations provide the required feedback to produce the coordination of belief, expectation, and regulation generally attributed

to social norms. Normative influence, expressed as the perception of common expectations within a domain of information, originates with individual perceptions. This drives the propagation of an emergent norm throughout the group as well as establish the boundaries of its domain.

This distilled and compact form of collective expectation is an effective vehicle for social curation of individual information. By compressing the diversity of individual experiences within a particular domain into a heuristic for expectation and uncertainty, norms and normative expectations curate the collective information of a group of individuals solely through interactions within the social network.

Furthermore, these heuristics are readily communicated—whether explicitly or implicitly—for comparison with (or in the absence of) individual experience. The curated information, in this heuristic form, suffices to encompass a collective "memory" of prior experience related to that domain by members of the social group.

## 5. Institutions and the Curation of Social Information

The standard definition of an institution, like those for norm and normative, has largely focused on the enforcement and regulation of social practices. Moreover, also like the concept of social norms, there is only a broad consensus as to what constitutes a social institution and its specific role in regulating group behavior [11,52,55,57,121,271]. Current accounts emphasize the practical or evolutionary effects of human cooperation, reciprocity, sanction, and various functional costs and benefits [55,60,67,78,109].

Whether couched in terms of promoting and enforcing rules of practice or as coordinating optimal strategic equilibria, institutions are considered centralizing and regulatory social entities through which cohesive practices of a group are maintained. In a sense, it is a logical consequence of viewing social norms as rules and/or strategies—rules require referees, so the coordination of socially normative practices requires a coordinating entity [11,74,85,133,272,273]. This framework, however, neither addresses nor defines the source or establishment of such conventions and institutions.

Mutually intelligible social interactions would require a preexisting or inherent commonality of such constitutive rules [34,39,55,57,60,61,66,78,272,274]. This presupposes some form of rudimentary or intrinsic normative institution as an innate attribute of human social cognition. The premise implies that social interaction depends on the prior existence of an institutional structure, but provides no clear causal pathway for institutional formation. Neither is there any clear rationale for acceptance, influence, or authority of such a regulatory social entity. Likewise, such explanations cannot address the divisions of the various institutions and the nature of those particular domains.

How would we even begin to evaluate the numerous distinct institutions that function concurrently within the same society? Furthermore, the process by which an optimal equilibrium or rule is determined remains an open question. Consider, instead, the function of social institutions as the high-level curation of collective information from which evaluative criteria are derived and by which to determine what might be considered optimal. Following from the processes of norm emergence and normative convergence described above, the establishment of a social institution represents the consolidation of pertinent information relating to domains of socially instrumental expectations.

The scope of an institution depends only on the identification of a domain of collective expectations to which they would be relevant. Rather than a functional necessity for promoting or enforcing coordinating rules to enable social interactions, social institutions become the consequence of the predicate coordination of expectations that arises from such interaction. Convergence of informative norms within a given domain identify the experientially validated bounds of the related expectations. Once determined, these constraints delineate common and collectively validated beliefs regarding that domain of information. In doing so, the domain is established as a socially pertinent and relevant node of information, which is validated by the coalescence of collective practices towards a stable (i.e., optimal) state.

Effectively, this stable state reflects the posterior coordination of information from which any equilibrium strategies or rules may be derived. An institution is the repository, not the arbiter, of collectively validated information and related practices. The basis of an institution then aligns to the curation of the collective and normative expectations that identify the bounds or limits of feasible equilibria within the identified domain of socially relevant information. Once these delimited ranges of potential equilibria coalesce, subsequent normative expectations within the bounds of those limits may take on institutional qualities in the traditional sense (e.g., proscriptive or prescriptive conventions).

In the case of institutions, the initial formation is the convergence of individual performances towards a collective pattern of normative expectations. The functional impetus for social institutions shifts from an a priori role in regulating performance towards an expression of the collectively evaluated bounds of optimal experiential outcomes. Institutionalized conventions and practices represent the social maintenance and retention for such experientially tested and validated information and expectations. The perceived regulatory function is the practical expression of this maintenance by retaining and communicating the parameterized convergence of that validation.

### 5.1. Curation and Instrumental Information

Whether culturally expressed as formal institutions or informal conventions, the establishment of these nominal constraints entails a posterior parametrization for the various expectations (e.g., belief, norm, collective, and normative) with which they are associated. Each expectation corresponds to reflexive interactions between individual and collective experiences, but the process itself promotes both coordination and assortative clustering of the underlying information. This information aggregates the collective expectations related to that domain by translating the underlying information into boundary constraints. These constraints are then available to the collective through the institutional repository of curated information, providing a socially vetted set of prior expectations (see [121,275,276]).

More importantly, this association between collective and normative expectations relates to the ability to render prior information instrumental for optimizing social strategies and practices. By elucidating the experientially supportable bounds of feasible expectations, curatorial social institutions provide the framework in which conventions, rules, or strategies may arise. Most notably, conventions and institutions frequently relate to domains associated with collective uncertainty and/or systemic risk. As discussed above, the convergence of normative expectations towards a stable posterior state elicits a sort of conceptual parameterization. While the normative expectation relates to the average collective expectation, the perception of risk or uncertainty derives from the expectation of deviation or variance from that average expectation (consider [277–279]).

We note a clear distinction here between the functional role of a social institution and the cultural and behavioral expression of that institution. The latter pertains to the behavioral practice and performance of the former. It is the social embodiment of collective behavioral practices through which the consequent expression of the equilibria is derived. The institution, then, is the emergent structure of the collective belief network, which sets the organizational structure of the social network itself.

The functional social institution facilitates the curation of pertinent information criteria that enable such instrumental expressions. Notably, specific cultural expressions of social institutions tend to be regulatory in nature, particularly for domains of social interaction relating to conflict resolution, risk, and ambiguity that occur with regularity [121,279,280]. Such socially critical domains, for which individual experience and expectation are less sufficient, are also where excessive variance in the bounding criteria could be detrimental.

Normative expectations for social interactions and performance, and institutional demarcation of their feasible optima, would encourage precise and unambiguous criteria of evaluation. The degree of that precision for a domain of interactions is proportional to the risk, whether individual or collective, associated with erroneous performance. The relationship of variance, which is both conceptually and technically the inverse of precision,

with expectation are dependent on their combined relationship with prior and posterior information [281].

Institutions act in a curatorial capacity for socially derived information and expectations of variance, and therefore reflect a collective classification mechanism for deriving and evaluating the optimality of expectations within a given domain of social interactions. This understanding likewise resolves what has otherwise been vaguely identified as "institutional memory"—memory implying the sentience of an inanimate entity, better explained as a clear product of information residing in some part with all constituents.

The behavioral expressions based on those optimal expectations would therefore manifest rule-like or strategically oriented structures, although remaining rooted in the optimization of expectations. Since the information from which that optimization is derived depends on the successive outcomes of individual and collective social interactions, the behavioral expressions of institutions are adaptive. Overly rigid institutional rules become infeasible or maladaptive when naturally accruing adjustments to underlying information and expectations exceed the allowable tolerance of the institutional parameters [282,283].

The processes of information and experience that collectively inform the system in its entirety do, however, imply a lag between stable state convergence and reestablishing the institutional optimization for the adjusted expectations. This can be seen in cases where the social experience of individual expectations and institutionally normative expectations diverge, with behaviorally expressed institutional expectations exhibiting conservative tendencies towards prior (rather than posterior) information and expectations.

By distinguishing between the functional and curatorial basis of institutions and their behavioral expressions [60,78,120–122,284–287], our formulation clarifies the dependency of institutions [11,55,199,271,284,288]. This resolves the ambiguity between viewing social institutions as either social entities or social facts away from contradictory proposals where institutional rules are intrinsically constructed or innate.

The status of social institutions becomes part of the overall framework within the process of transforming social experience, through information, into social expectation. Curatorial social institutions arise from the aggregation and validation of normative expectations, which in turn are themselves the consequence of the convergence of collective experience. This provides the conceptual framework through which cultural and behavioral expressions of norms, normative expectations, and their institutional forms may be enacted as both individual and collective behavior.

## 5.2. Social Information, Risk, and Efficiency

Throughout these discussions, we have suggested aspects of the underlying utility of the processes of socially embedded information—its estimation, verification, validation, and curation. We have only tangentially referred to the utility of social information as relating to individual and collective expectations, practice and performance, and mitigation of risk. To more concretely address both, we need to revisit our earlier observations about how, as socially situated agents, people habitually and unconsciously internalize a myriad of small practices as normal.

The processes of social information serve to filter and condense the vast amounts of data from individual experiences, while the collective validation and aggregation distills these expectations into curated repositories of prior information [266,266,270,289]. The remaining question is how does this process of normalization, culminating in numerous small and seemingly arbitrary or inconsequential practices, provide both individual and collective utility? In other words, what does being or recognizing "normal" actually accomplish?

The question can be partially addressed by looking at the underlying utility of information itself. The processes of norm estimation and the corresponding processes of normative parametrization constitute a collective search for informative priors to reliably project future expectations from past experiences. The outcome of that search establishes the expectation and range of variation by aggregating the individual and collective experiences within an

associated domain of belief. The utility of that aggregated information is in the ability to anticipate and adapt, which are inherently risk mitigation strategies [166,290–293].

Being able to anticipate a reasonable range of expectations for outcomes allows individuals and groups to make informed decisions in the assessment of possible strategies [290,294–296]. Similarly, adaptability or improvisation require some estimation of both the expected outcomes and the associated range of variability within those expectations (consider [297]). Each entails an evaluation of risk based on both the accuracy of the expectation and the likelihood of divergence from that expectation, even if the estimate is accurate. It is this ability to anticipate unknown outcomes efficiently and accurately and to estimate risks that promotes adaptation and innovation. The alternative—i.e., random experimentation or "trial and error" in the absence of information—is both costly and inefficient.

Less obvious is why the normalization of collective experiential information is expressed as an assemblage of disparate and particularized practices. That requires an exploration of the relationship between information and probability. The estimation of expectation, as described above, results in an estimation of the probability for an event or outcome. The goal is to maximize the accuracy of prediction and minimize the risk of an unexpected outcome. Shannon [298] described the relationship between probability and information by demonstrating that the likelihood of an outcome is inversely proportional to the amount of information required to minimize the uncertainty of its prediction.

Simply put, the occurrence of a rare event is more surprising than a common one and so the rare event yields more information. For common events, surprise is low and variance yields less overall risk to predicting outcomes. The expectation itself is enough to predict what is likely, and the variability of such common events is well-known. It has a minimal effect on shifting its expectation, since it is weighed against all other occurrences. The opposite holds true for rare events—accurate prediction is highly sensitive to variance (see [297,299]).

In the case of norms and normative expectations, the "event" to be predicted is the likelihood of an experience or belief—i.e., the expectation about an expectation. Finding that someone within your social network has a similar experience or belief should be less surprising than finding that they did not. Likewise, there should be a common expectation of just how different those experiences and beliefs ought to be before being recognized as dissimilar (i.e., surprising).

By normalizing and curating collective expectations, this risk of intolerable variance is strongly mitigated by and within a social network. Norms and normative expectations allow accurate estimation and prediction by minimizing the risk and effort associated with variance that exceeds energy-efficient conditions (see [300–302]). Establishing a common expectation (*norm*) and range of tolerable variance (*normative*) allows the interactions within a functioning social network to become a low-surprise environment. This reduces the amount of novel or surprising information, which reduces the effort required to process that information. Parsing those norms and normative expectations into particularized subsets of related domains of information (i.e., into diverse and specialized individual domains of practice) minimizes the effort needed to assess and determine a response to a situation or interaction.

The effect is that the social environment is optimized to minimizing the amount of information that needs to be processed during common interactions and practices [297]. Social norms, normative expectations, and institutions promote efficiency for individuals and the social collective. They do so by creating and maintaining an environment that minimizes the information required to be actively processed by its members. The utility of being and recognizing "normal" is a function of the efficiencies afforded by that low-surprise environment.

Social norms, and all related social constructs, exist to minimize surprise in the social environment and to maintain the working efficiencies of that environment.

### 6. Discussion

Definitions and concepts of social norms are primarily dependent on the particular area and interest of scholarly research [2,8,33,49,57,78,120,122,303,304]. While much of this work has been productive within its own scope, the broader questions regarding the evolutionary or causal sources of norms and normative influence have remained open questions. More importantly, existing models prove conceptually difficult to reconcile or apply outside their particular research areas due to incommensurate definitions and chains of causal inference.

Norms, normative influence, and social institutions have largely been viewed as related but discrete social entities, each operating at different scales of analysis. Moreover, norms and institutions are themselves generally partitioned and categorized by their particular domains of influence. These domains are distinguished by public or private spheres of interaction, interpersonal or individual modes of behavior, or differentiation of normative force. Institutions of governance or economics are treated as though separate and distinct from institutions related to family or religion, whereas norms are perceived as relatively unrelated subsets of rules emanating from those distinct institutions—the norms of fashion are treated as fundamentally and operationally distinct from the norms of commerce.

We have argued that a shared pool of information not only facilitates the sustained fitness of the collective and its individual members, but also mutual comprehension and predictability of individual actions and experiences. This allows for both global and individual optimization towards beneficial strategies and outcomes for the satisfaction of primal needs and social comity, thereby ensuring both the fitness and cohesion of the group. Effectively, the organizational and behavioral complexity resulting from the formation and evolution of normative institutions, as well as the consequent diversity of collective and individual behaviors, are byproducts of this process. They are particular manifestations of the information, filtered through individual and collective processes, and acquired from the unique experiences and environments of its constituents.

In essence, the broad behavioral patterns within and between individuals and groups (both past and present)—what is identified as culture—are the specific expressions and manifestations of the processes we have described above. Although anthropologists, sociologists, and others have traditionally defined culture in terms of such collective commonalities of traits, the source of both commonalities and differences between groups make such definitions problematic. As with definitions for social norms, an explanatory definition for culture has remained elusive.

If culture is defined by differential processing of information norms and their institutional expressions, however, then we can begin to better understand both how and why different cultural expressions evolve. Culture, then, is a bounded social and belief network of communities—whether bounded spatially, temporally, or by interaction—characterized by its collective processing of information norms and their resultant normative frameworks. In other words, a culture is the product of its communities finding their own collective "normal". Culture is the collection of normative institutions that arise from a particular population's validation and curation of its own historical information, which is derived from that population's unique collective experiences. That cultures adapt and evolve through time, or sometimes fail to do so, is explained by the constantly shifting landscapes of information and belief described by the processes we have outlined throughout this paper.

Similarly, we are saying that social norms and normative institutions are not discrete entities, but rather expressions of the same underlying set of cognitive processes. Those same processes generate the belief networks establishing the domains of norms or institutions. It is only the domain of influence and degree of associated risks that give the impression of distinct entities—i.e., norms of commerce differ from those of fashion only by the domain and risks. The underlying information process by which the norms and institutions take shape, however, are the same.



It is not merely that norms represent a form of social information, though. This fundamental role of norms in curating social information highlights the systemic risks of misguided or malicious manipulation of such norms or the information they represent. Recall that the underlying interactions of belief networks do not necessarily lead to verification of the truth of the information itself, but only the likelihood of belief. The same processes may also promote belief toward false information, and the overlapping interactions between belief networks would have a compounding effect. If norms—and consequently their dependent social entities—are not just mechanisms of social cohesion, then understanding their processes and effects becomes a critical issue in promoting and maintaining civil society.

Our core assertion is that it is the information that social norms represent that is the objective of social behavior, and ultimately the source of its adaptive advantages. Norms and the collective practices they promote are not byproducts of, or mechanisms for, the evolution of social behavior, but the foundations on which it rests. They are part of a process of information capture that allows collective and distributed acquisition, validation, and curation of experiential information. This subtle, yet fundamental, inversion of the causal trajectory of sociality reconciles several open questions surrounding normativity and institutions as well. Reconsidering information as a critical resource that is shared, validated, and curated through a distributed network of social actors realigns the purpose and adaptive fitness of sociality itself.

**Author Contributions:** All authors have contributed equally to the conceptualization and writing for this article. All authors have read and agreed to the published version of the manuscript.

**Funding:** This research received no external funding.

**Data Availability Statement:** No new data were created or analyzed in this study. Data sharing is not applicable to this article.

**Conflicts of Interest:** The authors declare no conflicts of interest.

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
