# Peer review of "The Behavior of Information: A Reconsideration of Social Norms"

_societies, doi:10.3390/soc13050111_

Round 1

Reviewer 1 Report

I was excited to review this manuscript when invited after reading the Abstract—general theory is an interest and what the authors signaled they were going to develop regarding social norms sounded interesting and ambitious. Therefore, the actual manuscript proved to be quite disappointing for the reasons I explain below. The authors gesture toward major arguments on norm development in the cognitive and cultural social sciences, but ultimately don’t engage with this argument accurately. Overall, the authors really do not propose a new theory, which is ostensibly the ambition.

1: The authors seem to be engaging with a very old theoretical paradigm of cultural socialization most identified with Talcott Parsons that imagines individuals as ‘dumping grounds’ for culture. In the way that they describe the irrationality if norm-following and seeking somehow trying to square seemingly irrational norms with some sort of group + individual functionality is misguided. The authors may find it helpful to read Jospeh Henrich’s The Secret of Our Success (2016). The key argument here is that because we increasingly depended on our groups to meet our needs as a species signaling that we are good members of our groups was a pressure on our evolutionary development in-and-of itself. That is, individuals who were good members of their groups—who signaled this by following the group’s norms, even if they appear irrational from the outside—were deemed to be good potential mates, because they are signaling their good standing in the group, and therefore less likely to be exiled or killed. There were moments where I thought the authors were going to engage with this key argument in the literature, but ultimately was unsatisfied.

2: The authors define many terms and concepts in the literature on norms and culture—beliefs, values, institutions, etc.—in a way that makes it seem that they are proposing new concepts. While we want to be precise about what we mean when we use these kinds of terms in developing our arguments, but there really is nothing new here. These terms have quite precise meanings in the cognitive and decision sciences. For a review, see Vaisey and Valentino 2018.  

3: The manuscript has not been prepared carefully. Sections 2 and 3 are the exact same text. This may seem as a minor flaw, but it is not—it is a signal of the care the authors put into their submission.

Reference

Vaisey, Stephen and Lauren Valentino. 2018. “Culture and Choice: Toward Integrating Cultural Sociology with the Judgment and Decision-Making Sciences.” Poetics 68:131-143.

Reviewer 2 Report

This paper deals with a topic of great importance for social studies. It is logically structured and comprehensible. This research deals with various aspects of social normativity.

One of the great virtues of the text is that it deals with a current and classic topic, but does so from a primarily psychological and informational perspective. This is not a problem, but I think it would be of great importance to show the position from which one is going to speak. That is, in the article there is no reference to great authors (Habermas, Weber, Luhmann, among others) linked to sociology who have contributed much to the subject of this paper. I understand that this is due, as I said, to the author's perspective. Well, according that I think these premises should be made very clear. At present it is intuited, but it is not clear.

On the other hand, in several places (for example on page 6 line 304, or on page 18 line 909) it seems that the term normal or normality is used and it seems that its meaning is colloquial. This should be avoided as it is confusing. The concepts of social norm and "the normal", in their colloquial use, are not comparable. Indeed, the social norm can be conceived as prior information of a macro nature, "the normal", in its colloquial use, is closer to individual and group conceptions (micro and meso). I think this idea is well reflected in this article on Goffman: https://doi.org/10.1111/0735-2751.00143. Therefore, I suggest the authors to eliminate the terms referring to colloquial use.

Third and finally, on page 8, line 380, it is stated that sociality promotes norms. This is interesting, but I think it could only be adjusted to human societies. As the authors also refer to biological phenomena (page 9), I humbly believe that they could adjust this to indicate that sociality, in socially undeveloped organisms, promotes regularities and in socially more complex organisms it promotes norms. This is better explained in this text: https://doi.org/10.1016/j.biosystems.2021.104552  

In summary, the paper is of great interest to the scientific community, from an informational and psychological perspective. As is often the case in theoretical papers, the reader may always have the feeling that things are missing. In this case, for future research, I suggest that the author focus on a specific topic and develop it extensively.

Regarding the structure of the article, I recommend several changes: 

1. Section two and three have the same title.

2. Section four could be the discussion of the text, since it is where the core of the article is discussed.

3. The current discussion section could be the conclusions.

Reviewer 3 Report

Review for Societies 

This interesting paper on social norms and how they relate to communication fits well into the scope of the journal.

At the same time, there are some issues the authors should address before publication. In particular, there are a number of claims, some rather bold, that are not backed up with any reference. Other paragraphs are rather unspecific. The authors should improve the clarity of their arguments, avoid overly bold statements, and strengthen the links to the literature. There are also large parts of text lacking any reference. This must be fixed as a precondition for publication. I am rather ambivalent between reject and major revisions, then decided for major revisions. The authors must revise their text very conscientiously, convincing the editors and reviewers that they know where their arguments come from.

The introduction should be backed up with suitable references. There is plenty of work on the evolution of norms. The authors should not only refer to the notion in the first two paragraphs of their introduction but also add links to suitable previous work.

For example: 

Norms evolving spontaneously:

Young, H. P. (2015). The evolution of social norms. economics, 7(1), 359-387.

Norms evolving due to policy intervention:

Berger, J., Efferson, C., & Vogt, S. (2021). Tipping pro-environmental norm diffusion at scale: opportunities and limitations. Behavioural public policy, 1-26.

In the following paragraph, they mention norms, conventions, institutions. Here too, the authors need to add suitable references.

For example:

Bicchieri, C, 2005, The Grammar of Society.

Coleman, 1990, Foundations of Social Theory.

The authors cite some work in references 1-5, but the connection between the text and the references should be much stronger, making it clear which scholar contributed what to the knowledge the authors draw on.

Likewise, the processes by which norms emerge (barring external imposition) and 118 how they evolve remain open questions [3,36,37].”

I am not convinced that we do not know anything about the questions. The authors should be a bit more precise here. What do we know, what exactly is still unknown? There is quite some theory and evidence on the diffusion of norms. Bicchieri provides a nice framework in 

Bicchieri, 2016, norms in the wild.

Empirical studies on the diffusion of norms include:

Berger, J. (2021). Social tipping interventions can promote the diffusion or decay of sustainable consumption norms in the field. Evidence from a quasi-experimental intervention study. Sustainability, 13(6), 3529.

Berger, J., Efferson, C., & Vogt, S. (2021). Tipping pro-environmental norm diffusion at scale: opportunities and limitations. Behavioural public policy, 1-26.

Centola, D., Becker, J., Brackbill, D., & Baronchelli, A. (2018). Experimental evidence for tipping points in social convention. Science, 360(6393), 1116-1119.

Centola, D., & Baronchelli, A. (2015). The spontaneous emergence of conventions: An experimental study of cultural evolution. Proceedings of the National Academy of Sciences, 112(7), 1989-1994.

Nyborg, K. (2020). No man is an island: social coordination and the environment. Environmental and Resource Economics, 76, 177-193.

The authors should acknowledge the literature and more precisely specify what we do and don’t know about the emergence of norms.

Ostrom, E. (2000). Collective action and the evolution of social norms. Journal of economic perspectives, 14(3), 137-158.

Young, H. P. (2015). The evolution of social norms. economics, 7(1), 359-387.

“The adherence and enforcement of norms can be highly variable – 133 at times the strictest of norms may be violated without significant sanction, while other 134 insignificant transgressions can have profound social consequences. ” p. 3
Is this so? If yes, please add suitable references. This is a very bold statement that needs to be supported with evidence.

“Such cooperative and fairness norms make sense for both individuals and the group 163 from an evolutionary perspective, and a substantial body of research has demonstrated 164 just that utility [22,55,64–70]. “ P: 4

Quite often, so-called cooperation norms remove the tension between group and individual. The authors should address this widespread notion. They also might wish to refer to sanctions as a means of creating compliance.

Suitable literature is the following:

Coleman 1990, foundations of social theory

Fehr, Gächter, 2002, altruistic punishment in humans, nature

Berger, Hevenstone, 2016, norm enforcement in the city revisited, rationality and society

Page 5: Large text segments without references. Where does all of this come from? The notion of shared expectations, for example?

This explanation neglects the origins 399 of that social environment and the effect of individual and collective adaptations as a 400 generative interaction between agency and environment. p. 9

What, exactly, is meant by “the environment” here? Please explain, again, referring to the literature.

“Norms emerge as the perceived central tendency within conceptual domains identified by that information. What have been described as norms and normative institutions should instead be considered the performances that emerge around that framework for retention and maintenance of information.” p 9.

Interesting. But: Where does this come from? Rationale, evidence, references?

Beginning of section 4.2. Again a lot of text without any reference. 

The same holds for almost the entire section 5. References are a must for a scientific text.

Round 2

Reviewer 3 Report

The authors have strengthened the link to the literature as suggested. Although I think that some parts of the paper (including the clarity of some arguments) could still be improved, the paper is now ready for publication in Societies.